# Paraffin-enabled graphene transfer

Wei Sun Leong [1], Haozhe Wang [1], Jingjie Yeo[2,3,4], Francisco J. Martin-Martinez[3], Ahmad Zubair [1], Pin-Chun Shen[1], Yunwei Mao[5], Tomas Palacios[1], Markus J. Buehler[3], Jin-Yong Hong[1,6] & Jing Kong[1]

The performance and reliability of large-area graphene grown by chemical vapor deposition are often limited by the presence of wrinkles and the transfer-process-induced polymer residue. Here, we report a transfer approach using paraffin as a support layer, whose thermal properties, low chemical reactivity and non-covalent affinity to graphene enable transfer of wrinkle-reduced and clean large-area graphene. The paraffin-transferred graphene has smooth morphology and high electrical reliability with uniform sheet resistance with ~1% deviation over a centimeter-scale area. Electronic devices fabricated on such smooth graphene exhibit electrical performance approaching that of intrinsic graphene with small Dirac points and high carrier mobility (hole mobility = 14,215 cm$^2$ V$^{-1}$ s$^{-1}$; electron mobility = 7438 cm$^2$ V$^{-1}$ s$^{-1}$), without the need of further annealing treatment. The paraffin-enabled transfer process could open realms for the development of high-performance ubiquitous electronics based on large-area two-dimensional materials.

[1] Department of Electrical Engineering and Computer Science, Massachusetts Institute of Technology, Cambridge, MA 02139, USA. [2] Department of Biomedical Engineering, Tufts University, Medford, MA 02155, USA. [3] Laboratory for Atomistic and Molecular Mechanics, Massachusetts Institute of Technology, Cambridge, MA 02139, USA. [4] Institute of High Performance Computing, A*STAR, 1 Fusionopolis Way, Singapore 138632, Singapore. [5] Department of Mechanical Engineering, Massachusetts Institute of Technology, Cambridge, MA 02139, USA. [6] Carbon Industry Frontier Research Center, Korea Research Institute of Chemical Technology, Daejeon 34114, Republic of Korea. These authors contributed equally: Wei Sun Leong, Haozhe Wang. Correspondence and requests for materials should be addressed to J.-Y.H. (email: jyhong@krict.re.kr) or to J.K. (email: jingkong@mit.edu)

Charge carriers in graphene exhibit ultrahigh mobility owing to their zero-mass nature in the linear E−k dispersion at low energy, where the conduction and valence bands meet at the Dirac points[1]. Although mobility values as high as 200,000 cm$^2$ V$^{-1}$ s$^{-1}$ have been reported for freely suspended graphene exfoliated from bulk crystal at 5 K[2], the mobility values reported for large-area graphene is several orders of magnitude lower, regardless of the growth method or substrate employed[3–7]. This significant decrease in carrier mobility in large-area graphene can be attributed to four factors: (1) polycrystalline nature of graphene, (2) effect of surrounding medium, (3) contamination from transfer support layer, and (4) wrinkles present in graphene. Any of these factors can reduce the carrier mean free path in graphene, which could act as a source of extrinsic scattering, thus limiting the carrier mobility[8]. To address the first factor, there are considerable advances in the synthesis of wafer-scale single-crystal graphene domains[9,10], with the latest achievement of 1.5-inch-large graphene monolayer synthesized on Cu–Ni alloys[11]. As for the second factor, the effects of the surrounding medium on graphene can be minimized by fully encapsulating the graphene layer within insulating substrates through van der Waals assembly approach[12]. For instance, a mobility value of 110,000 cm$^2$ V$^{-1}$ s$^{-1}$ at 1.6 K was reported for an as-synthesized chemical vapor deposition (CVD) graphene flake sandwiched between hexagonal boron nitride (h-BN) flakes[13]. Furthermore, to fabricate graphene-based functional electronics, graphene must be transferred from the growth substrate to a destination substrate. Since graphene is only one atom thick, a support layer is required during the transfer process to prevent cracks from appearing and propagating in the graphene film. To date, after one decade of research efforts, the most widely used support layer for graphene transfer is still polymethylmethacrylate (PMMA). However, the PMMA-assisted transfer process engenders two major issues that severely degrade the carrier mobility in graphene: polymer contamination and graphene wrinkling (third and fourth factor mentioned above). To overcome these two issues, support layers made of other polymers and organic molecules have been attempted[14–21], but none of these solutions fully solved both issues.

We have shown earlier that paraffin can be used as a flexible substrate to transfer graphene onto it[22]. Here, we propose to use paraffin as a transfer support layer based on two rationales: (1) paraffin is an alkane with a simple unreactive chemical structure and (2) it has a high thermal expansion coefficient. Wrinkles form in graphene either natively during the growth process[23] (both CVD and epitaxial graphene) or during the graphene transfer process, where water (including vapor) can be trapped between the graphene and the destination substrate[24]. To address the graphene wrinkling issue, we scoop up the paraffin-supported graphene layer with a destination substrate from deionized water at a higher temperature of 40 °C, rather than the usual room temperature. At 40 °C, the paraffin support layer undergoes thermal expansion due to the small amount of heat from the water[25]. This expansion can stretch the graphene film underneath and thus effectively reduces wrinkles in graphene.

Our paraffin-enabled transfer technique can concurrently addresses both support layer contamination and wrinkling in graphene, enabling transfers of large-area graphene with homogeneous and enhanced electrical properties. Compared to PMMA, paraffin is found to leave much less residue on graphene, as it does not contain carbonyl (C=O) functional groups that react with electrophiles or nucleophiles. Moreover, paraffin radicals are not expected to form covalent bonds with graphene. Our density functional theory (DFT) calculations confirm that paraffin has lower adsorption energy with graphene compared to PMMA, which supports the effective curtailing of contamination

observed in our paraffin-transferred graphene. Through atomic force microscopy (AFM) and Raman studies, we confirm that paraffin-assisted transfers substantially reduce contamination and wrinkling in graphene compared to PMMA, and moreover, the paraffin-transferred graphene has near-intrinsic levels of doping and strain. Consequently, field-effect transistors fabricated on paraffin-transferred graphene exhibit near-zero Dirac voltage and the electron mobility that is ~fourfold higher than that of our PMMA-transferred graphene. Our paraffin-enabled transfer technique presented in this work offers a solution for the development of high-performance large-area graphene-based electronics by minimizing the charge carrier scattering centers in graphene.

## Results

**Paraffin-enabled transfer technique**. Our paraffin-enabled graphene transfer technique is straightforward as illustrated in Fig. 1a. In this work, CVD monolayer graphene grown on Cu foil was used because graphene synthesized in this manner is the most widely used source of large-area graphene in state-of-the-art research and industrial development, despite being polycrystalline and having lower mobility values. Details on the process of CVD graphene synthesis are available in our previous work[26]. As can be seen in Fig. 1a, a layer of paraffin was spin-coated first on the synthesized graphene to serve as a transfer support layer, followed by etching of the Cu growth substrate (see Methods for details). The paraffin-supported graphene was then rinsed with deionized water multiple times at room temperature. Subsequently, the sample was transferred onto 40 °C deionized water and kept at the same temperature for 1 h, where the paraffin layer remained at a solid state (Fig. 1c). This step facilitates isotropic thermal expansion in the paraffin layer following equation:

$$\frac{L_f - L_i}{L_i} = \alpha(T_f - T_i), \quad (1)$$

where $L_f$ and $L_i$ denote the final and initial lengths of a material; $\alpha$ is the linear thermal expansion coefficient of a material ($\alpha_{PMMA} = 70$ μK$^{-1}$; $\alpha_{paraffin} = 160$ μK$^{-1}$)[25]; and $T_f$ and $T_i$ are the final and initial temperatures of a material, respectively. Thermal expansion of paraffin translated to tensile strain on the graphene film underneath, which stretched and diminished wrinkles in the graphene film (Fig. 1b). The paraffin-supported graphene sample was then scooped up with a destination substrate from the deionized water at 40 °C and baked in an oven at 40 °C for more than 24 h to minimize the residual water. Finally, the paraffin layer was removed, leaving a graphene monolayer on the destination substrate.

**Homogeneous electrical properties**. We observed homogeneous and improved electrical properties in the polycrystalline CVD graphene prepared with our paraffin-enabled transfer technique. To determine the electrical properties, CVD graphene was cut into $15 \times 15$ mm$^2$ squared pieces and transferred onto Si/SiO$_2$ substrates using the paraffin support layer. Four small contact regions were then made by pressing a thin slice of indium wire with metal tweezers at the corners of each squared graphene film (contact length is less than 7% of the side length of the graphene film). Subsequently, resistivity and Hall measurements were made on each paraffin-transferred graphene film in the presence of 2800 Gauss magnetic field to extract its sheet resistance, carrier concentration, and Hall mobility values. During the Hall measurements, the magnetic field was applied by attaching a piece of magnet underneath the sample with desired polarity. For comparison, PMMA-transferred graphene film on Si/SiO$_2$ substrates were also fabricated with the same dimensions (see Methods for

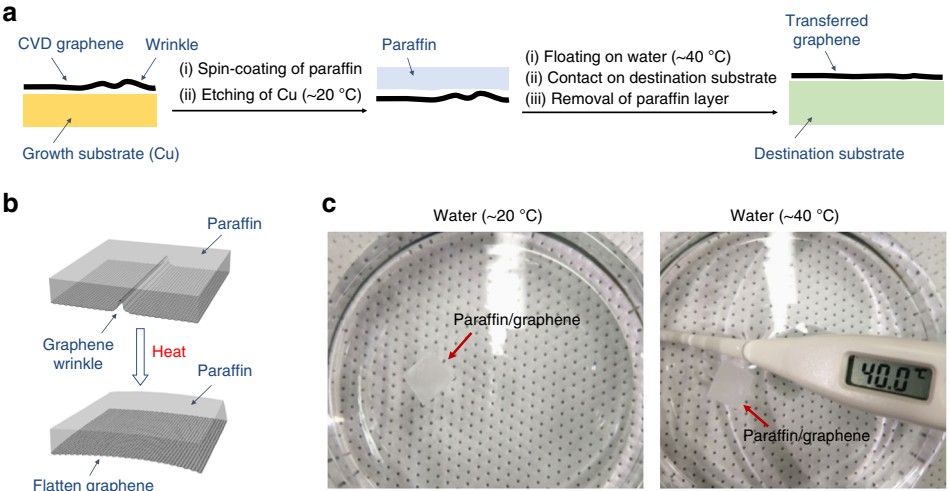

**Fig. 1** Our paraffin-enabled graphene transfer method. **a** Schematics showing the process of paraffin-assisted graphene transfer. **b** Schematics showing the effect of paraffin's thermal expansion on graphene wrinkle. **c** Photographs of a typical paraffin-supported graphene film floated on water at different temperatures as indicated confirming that the paraffin layer is still in solid state at ~40 °C

details) and measured. To avoid run-to-run variations, the graphene used in this study was synthesized on the same Cu foil in the same CVD growth cycle and transferred on $SiO_2$/Si substrates that were diced from the same $SiO_2$/Si wafer.

In Fig. 2a, we compared the Hall measurement results of 12 paraffin- and PMMA-transferred graphene films (6 for each). The average Hall mobility of paraffin-transferred graphene was $4365 \pm 347$ cm$^2$ V$^{-1}$ s$^{-1}$, which was 2.6 times higher than the PMMA-transferred graphene ($1698 \pm 377$ cm$^2$ V$^{-1}$ s$^{-1}$). Furthermore, the carrier concentration of paraffin-transferred graphene ranged from 0.9 to $2.6 \times 10^{12}$ cm$^{-2}$, which was lower than that of PMMA (7.2 to $12 \times 10^{12}$ cm$^{-2}$). The lower carrier concentration indicated that the paraffin did not dope graphene as heavily as the PMMA. We note that no annealing was performed on any transferred graphene in this study. For the same group of graphene samples, sheet resistance was also extracted using the van der Pauw method and plotted in Fig. 2b. The sheet resistance of paraffin-transferred graphene was considerably lower with a narrower distribution in comparison with that of PMMA-transferred graphene. To gain further insights into the spatial distribution of sheet resistances across these large-area graphene films transferred with different support layers, 800 sheet resistance values were measured using a four-point probe tool over graphene films with dimensions of $40 \times 20$ mm$^2$, with 1 mm step size in both the $x$- and $y$-directions. The spatial sheet resistance maps of both PMMA- and paraffin-transferred graphene are plotted in Fig. 2c, d, respectively, with the same color intensity bar. For PMMA-transferred graphene, the measured sheet resistances varied broadly from 446 to 916 Ω per square (average = $600 \pm 95$ Ω per square), with random areas showing relatively high average sheet resistances that could be due to damage on graphene induced by the transfer process, including minor cracks, wrinkles, and clustering of PMMA residues. In contrast, the paraffin-transferred graphene film exhibited lower sheet resistances with a much narrower deviation (average = $507 \pm 17$ Ω per square) that were distributed homogenously for the same sample size, with slightly increased sheet resistances along the edges of the graphene film. In short, the results indicated that the paraffin-transferred graphene had lower doping levels, smaller and more homogeneous sheet resistance values compared to the PMMA-transferred graphene, confirming high electrical reliability of paraffin-transferred graphene. We attribute the unusual relationship between carrier concentration

and sheet resistance to the damage on graphene induced by the PMMA transfer process[14].

**Enhanced electrical performance**. To examine the electrical performance of devices fabricated with paraffin-transferred graphene, more than 100 graphene back-gated field-effect transistors (FETs) were fabricated on Si/SiO$_2$ substrates and tested (Fig. 3a). Similarly, the graphene used in this study was synthesized on the same Cu foil in the identical CVD growth cycle for fair comparisons. Details on the processes to fabricate these FETs can be found in the Methods section. No annealing was performed on any FETs prior to back-gate measurements[27]. The electrical measurements were conducted at room temperature in air. In Fig. 3b, we compared the transfer characteristics of two typical field-effect transistors fabricated with PMMA- and paraffin-transferred graphene. From the transfer characteristics, the peak field-effect mobility, $\mu_{FE}$ of a graphene FET can be calculated via equation:

$$\mu_{FE} = \frac{1}{C_{ox} V_{DS}} \times \frac{L_{ch}}{W_{ch}} \times \frac{\Delta I_{DS}}{\Delta V_G}, \qquad (2)$$

where $C_{ox}$ is the gate capacitance ($1.21 \times 10^{-8}$ F cm$^{-2}$ for 285 nm thick SiO$_2$); $L_{ch}$ and $W_{ch}$ the channel length and width; $I_{DS}$ the drain-to-source current; $V_{DS}$ the drain voltage; and $V_G$ the back-gate voltage. For the typical transistor fabricated on paraffin-transferred graphene in Fig. 3b, the extracted hole and electron mobilities were 14,215 and 7,438 cm$^2$ V$^{-1}$ s$^{-1}$, while for the PMMA-transferred graphene, the extracted hole and electron mobilities were much lower, at 3,719 and 1,653 cm$^2$ V$^{-1}$ s$^{-1}$, respectively. Transferring graphene with paraffin greatly enhanced the effective hole and electron mobilities by 3.8 and 4.5 times, respectively, due to reduced charge carrier scattering centers in the paraffin-transferred graphene. Figure 3c compares the electron mobility distribution of 140 graphene FETs fabricated with PMMA- and paraffin-assisted transfer process. The electron mobility of paraffin-transferred graphene was four times higher than that of PMMA, regardless of the device channel length. It is worth noting that all FETs fabricated on paraffin-transferred graphene in this study were contaminated with PMMA once during the device fabrication processes[28].

Remarkably, the back-gate measurements also indicated that paraffin did not induce significant doping effects on graphene,

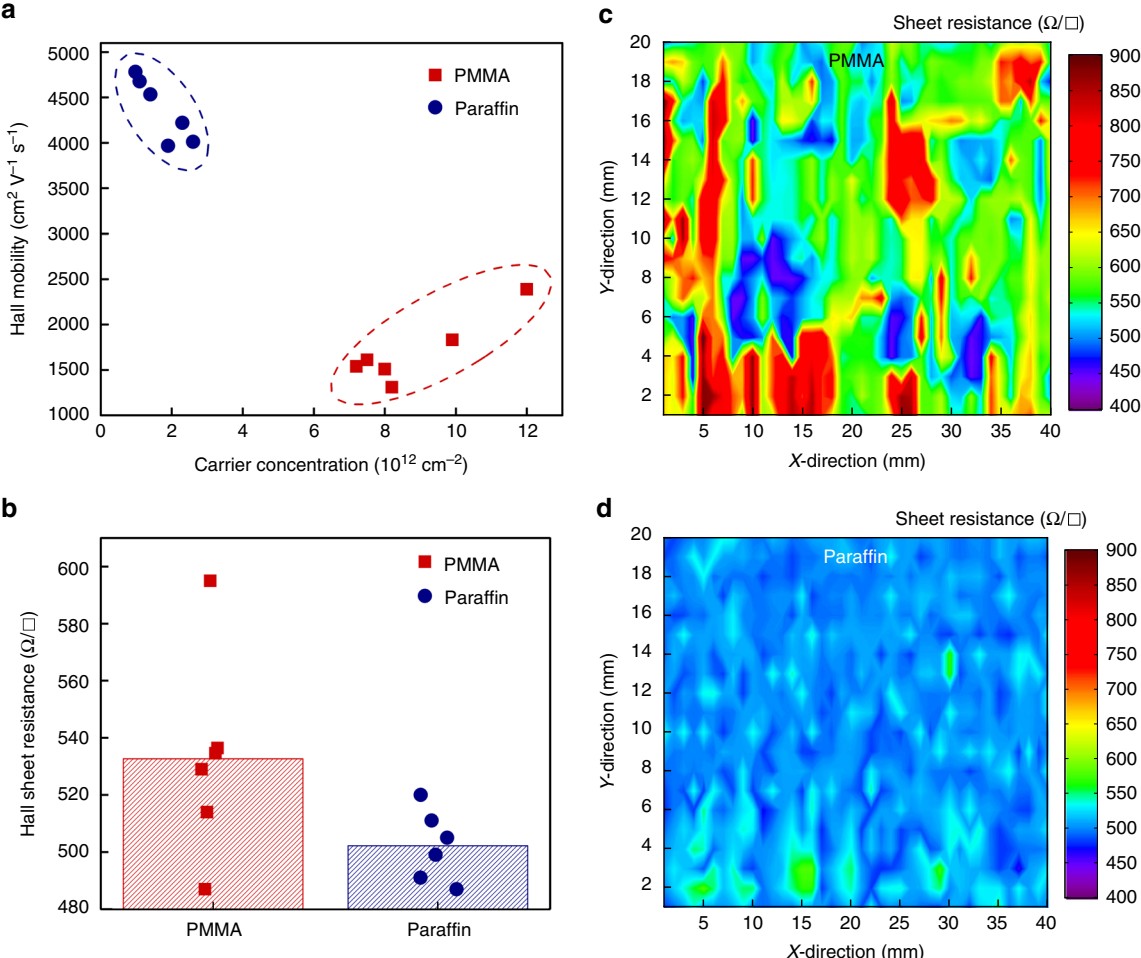

**Fig. 2** Electrical properties comparison of graphene transferred with PMMA and paraffin support layers. **a** Room-temperature hall mobility versus carrier concentration of graphene films transferred with different support layers as indicated. Each data were extracted from a $15 \times 15$ mm$^2$ graphene transferred on Si/SiO$_2$ substrate using Hall measurement in the presence of 2800 Gauss magnetic field. **b** Sheet resistance distribution of graphene films transferred with different support layers, obtained via Hall measurement. **c, d** Spatial sheet resistance maps of a graphene film transferred with **c** PMMA and **d** paraffin support layers. Eight hundred sheet resistance values were measured using four-point probe measurement over a graphene area of $40 \times 20$ mm$^2$, with 1 mm step size in both $x$- and $y$-directions. The paraffin-transferred graphene film exhibits much lower and homogenous sheet resistance than that of PMMA. In figure **a**, **b**, blue-color round-shaped symbols represent paraffin-transferred graphene, and red-color square-shaped symbols represent PMMA-transferred graphene

consistent with the Hall measurements in Fig. 2a. In Fig. 3b, the Dirac voltage ($V_{Dirac}$) of the device fabricated on paraffin-transferred graphene was much smaller and closer to zero compared to that of PMMA. The $V_{Dirac}$ of a graphene device is the specific back-gate voltage where the device operates with minimum conductivity, corresponding to the Dirac point or charge neutrality point where the graphene film's conduction and valence bands meet, and intrinsic graphene exhibits minimum conductivity when gate voltage equals to zero. Hence, $V_{Dirac}$ is a good indicator showing how intrinsic a graphene film is. Figure 3d compares the $V_{Dirac}$ distribution of 100 graphene FETs fabricated with both PMMA- and paraffin-enabled transfer process. The $V_{Dirac}$ of PMMA-transferred graphene had a spread from 24 to 56 V with a mean value of 38.7 V, while that of paraffin-transferred graphene had a much narrower distribution between 4 and 12 V with a mean value of 7.4 V, which is five times smaller than the PMMA-transferred graphene. The lower mean value for $V_{Dirac}$ signified that the paraffin-transferred graphene exhibited electrical properties that were closer to intrinsic graphene compared to that of PMMA, although the paraffin-transferred

graphene was also contaminated by some PMMA residues during the device fabrication process. The improved process of transferring graphene might play a vital role here in determining the electrical performance of large-area graphene devices, even though the device fabrication process was not optimized to prevent contamination by PMMA residues.

**Materials properties approaching to the intrinsic graphene**. By characterizing the material properties, we showed that our paraffin-transferred graphene exhibited doping and strain properties approaching that of intrinsic graphene. In Supplementary Fig. 1, we plotted the X-ray photoelectron spectroscopy (XPS) spectrum of PMMA- and paraffin-supported graphene transferred on Si/SiO$_2$ substrates. As can be seen, our paraffin-transferred graphene sample has no C=O and O=C–OH peaks, and the C–OH peak intensity is relatively weak, indicating lower levels of polymer contamination. Figure 4a, b depict the typical atomic force microscopy (AFM) height profile images of CVD graphene transferred on Si/SiO$_2$ substrate using PMMA and paraffin support layers, respectively. We note that all graphene samples were

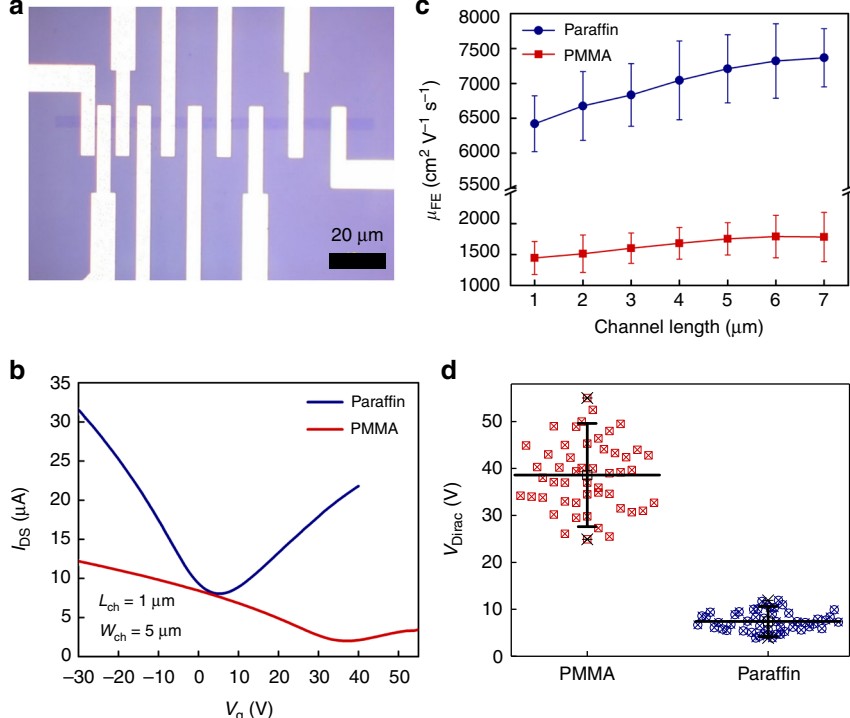

**Fig. 3** Back-gate electrical measurement results of PMMA- and paraffin-transferred graphene devices. **a** Optical image of a typical graphene field-effect transistors (FETs) array with increasing channel length. **b** Transfer characteristics comparison of two field-effect transistors fabricated with PMMA- and paraffin-transferred graphene. The Dirac voltage of device fabricated on paraffin-transferred graphene is much smaller and closer to zero. **c** Two-terminal field-effect (electron) mobility distribution of graphene FETs fabricated with PMMA- and paraffin-enabled transfer process as a function of channel length. Each average value was extracted from 10 graphene FETs. **d** Dirac voltage distribution of 100 graphene FETs fabricated with PMMA- and paraffin-assisted transfer. In **b**–**d**, error bars indicate standard deviations, blue-color round-shaped symbols represent paraffin-transferred graphene, and red-color square-shaped symbols represent PMMA-transferred graphene

not annealed after the transfer process. Compared to PMMA-transferred graphene, paraffin-transferred graphene had a much cleaner surface (Fig. 4b and Supplementary Fig. 2). Besides fewer polymer residues, paraffin-transferred graphene substantially reduced wrinkles compared to the PMMA-transferred graphene. In Fig. 4b, over a graphene area of $20 \times 20 \ \mu m^2$, there were very few wrinkles measuring more than $10 \ \mu m$ in length. More surprisingly, smaller wrinkles in size (both height and width) are not observable on the graphene surface. For the removal of graphene wrinkles, we inferred that the paraffin support layer could effectively smoothen these wrinkles through thermal expansion and releasing the compressive strain generated in graphene during the cooling step in the CVD growth process. To ascertain this inference, we collected 3600 Raman spectra from each type of the transferred graphene and analyzed the strain-doping relationship. Supplementary Fig. 3 shows the typical Raman spectrum of PMMA- and paraffin-transferred graphene on $Si/SiO_2$ substrate. The characteristic D band peak, typical in the presence of defects, was absent in paraffin-transferred graphene, indicating the transferred graphene was of high quality. In Fig. 4c, we plot a correlation map of G peak position ($\omega_G$), 2D peak position ($\omega_{2D}$), and 2D peak's full width at half maximum ($\Gamma_{2D}$) for the PMMA-transferred graphene. The black circle represents the G and 2D peak positions of an intrinsic graphene, where graphene has neither doping nor strain ($\omega_G = 1582 \ cm^{-1}$; $\omega_{2D} = 2670 \ cm^{-1}$)[29]. Similar to previous reports, our PMMA-transferred CVD monolayer graphene on $Si/SiO_2$ substrate experienced compressive strain and p-doping[30]. In contrast, our paraffin-transferred CVD monolayer graphene on $Si/SiO_2$ substrate experienced almost zero strain and very weak p-doping, approaching that of intrinsic graphene (Fig. 4d).

**Fundamental mechanisms of paraffin as a support layer**. Both electrical measurements and materials characterizations strongly suggested that the paraffin support layer did not interact strongly with graphene, and hence, we observed almost no contamination and doping effect on paraffin-transferred graphene. To unveil the fundamental chemical mechanisms underlying the distinct contamination level in PMMA- and paraffin-transferred graphene, we performed DFT calculations to optimize the geometries and to determine conceptual DFT chemical reactivity descriptors for both PMMA and paraffin chemical structures (Fig. 5a, b). Thus, the dual descriptor of the Fukui function and the chemical hardness were calculated for PMMA dimers and a short chain of paraffin. From Fig. 5c, d, the dual descriptor of the Fukui function for the PMMA dimer indicated that the carbonyl groups were very reactive compared to the rest of the molecule, whereas the reactivity of paraffin was generally equivalent throughout the molecule. These concentrations of local reactivity in the carbonyl functional groups increase the ability of PMMA to undergo chemical reactions in that part of the molecule. Furthermore, the chemical hardness for the PMMA dimer was $88.07 \ kcal \ mol^{-1}$, in contrast to the much higher chemical hardness of $157.04 \ kcal \ mol^{-1}$ for paraffin. This indicates a much higher resistance of paraffin to get involved in chemical reactions that imply any transfer of electrons. Taken together, these results on the local and global chemical reactivity demonstrated that PMMA was significantly more likely to react in the presence of any eventual chemically reactive species. Our DFT calculations further indicate that the adsorption energies indicated that PMMA adsorbed more strongly to the surface of graphene with an adsorption energy that was $1.56 \ kcal \ mol^{-1}$ higher than that of paraffin. As shown in Fig. 6a, formation of a paraffin-graphene interface is mostly driven by non-covalent interactions and no covalent bonds

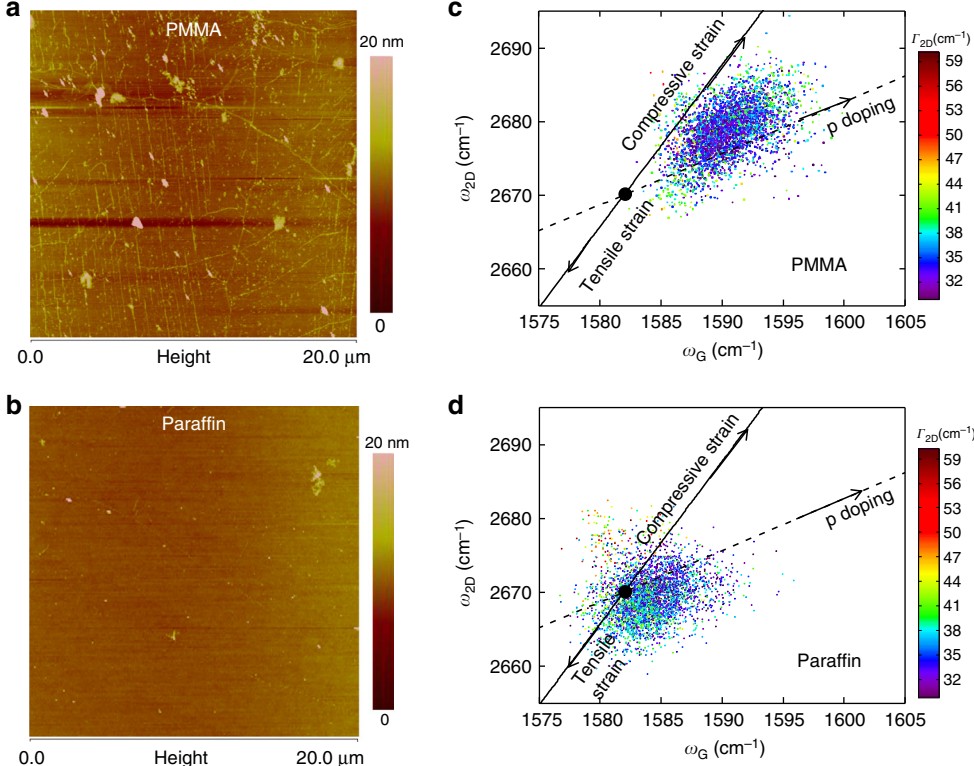

**Fig. 4** Materials characterization of the PMMA- and paraffin-transferred graphene on Si/SiO₂ substrate. **a, b** Typical AFM height profile images of graphene film transferred with **a** PMMA and **b** paraffin support layers. **c, d** Correlation map of the Raman G and 2D peak positions of graphene transferred with **c** PMMA and **d** paraffin support layers. A total of 3600 Raman spectra were taken from each type of transferred graphene and the corresponding G peak position ($\omega_G$), 2D peak position ($\omega_{2D}$), and 2D peak's full width at half maximum ($\Gamma_{2D}$) were extracted. The black circle represents the G and 2D peak positions of an intrinsic graphene, where graphene has neither doping nor strain

would be formed between paraffin and graphene, even in the presence of a vacancy in graphene. In contrast, PMMA radicals bond covalently to the defective graphene, even if the carbon atoms surrounding the vacancy defect rearranged themselves to form more stable 5- and 8-membered rings (Fig. 6a), as verified by the shared highest occupied molecular orbitals (HOMO) and lowest unoccupied molecular orbitals (LUMO) (Fig. 6b). All these calculations support the idea that the appearance of large amounts of PMMA contaminants on the PMMA-transferred graphene was likely due to the combined effects of stronger non-covalent interaction between PMMA and graphene, higher reactivity of PMMA, and possible covalent bonding between PMMA radicals and the vacancy defects in graphene at elevated temperatures. In sharp contrast, paraffin's chemical properties of lower reactivity, lower non-covalent affinity to graphene, and lesser likelihood of forming stable radicals that lead to covalent bonding, were remarkably beneficial for producing cleaner, large-area graphene sheets.

## Discussion

In summary, the technique we developed enables transfer of large-area graphene with almost intrinsic properties preserved. Our paraffin-transferred graphene exhibits homogeneous and improved electrical properties as a result of reduced support layer contamination and wrinkle compared to the PMMA-transferred graphene. Specifically, lower levels of polymer contamination can be attributed to paraffin's low chemical reactivity and low non-covalent affinity to graphene, as confirmed by DFT calculations. Reduced wrinkles in graphene, on the other hand, is due to the use of heat during the transfer process to induce thermal expansion in the paraffin support layer. Through electrical measurements and Raman studies, we confirm that paraffin-

transferred graphene experiences very weak doping and almost zero strain, approaching that of intrinsic graphene. We anticipate the paraffin-enabled transfer approach can be applied to other large-area two-dimensional materials in addressing the challenges of support layer contamination and wrinkle propagation. Considering the established paraffin coating technology (e.g. on papers, cars, floors etc.), we envision the integration of paraffin support layer in industrial scale production of wrinkle-free, pristine two-dimensional materials in near future.

## Methods

**Paraffin-supported graphene transfer**. To prepare the paraffin solution, paraffin pastilles were purchased from Sigma-Aldrich 18634 and kept in an 80 °C oven. Melting point of the paraffin used in this work was measured to be ~45 °C (Supplementary Fig. 4). To fabricate a paraffin-supported graphene sample, paraffin solution kept in an 80 °C oven was dropped on a sample (i.e. CVD graphene synthesized on Cu foil), which solidified in less than 1 min at room temperature. The paraffin-covered graphene on Cu foil sample was then put on a spin coater. Heat gun was used to blow on the sample making the paraffin melts again and the substrate temperature was measured to be ~50 °C using a thermocouple. Once the paraffin on sample turns into liquid form, the sample was spun at 1000 rpm for 2 min, with continuous heat gun blowing. The thickness of spin-coated paraffin layer used in this work was measured to be 20 μm (Supplementary Fig. 5). After spin-coating, the sample was floated on Cu etchant (Copper Etchant TFB, Transene) for 30 min to remove the Cu growth substrate. The paraffin-supported graphene film was then rinsed with deionized water multiple times at room temperature. Subsequently, the sample was transferred onto 40 °C deionized water surface and kept at the same temperature for at least 1 h. The destination substrate (i.e. Si/SiO₂ substrate) was then used to scoop the paraffin-supported graphene sample from the 40 °C deionized water from only one end. After that, the sample was baked at 40 °C in an oven for more than 24 h to minimize the residual water. In the event of incomplete removal of residual water, minor cracks randomly generated in the graphene (Supplementary Fig. 6). In addition, paraffin at 40 °C is rather fragile to handle and nitrogen gas blowing is not recommended. Finally, the paraffin support layer was removed using hexane solution for 2 h, leaving monolayer graphene on the destination substrate.

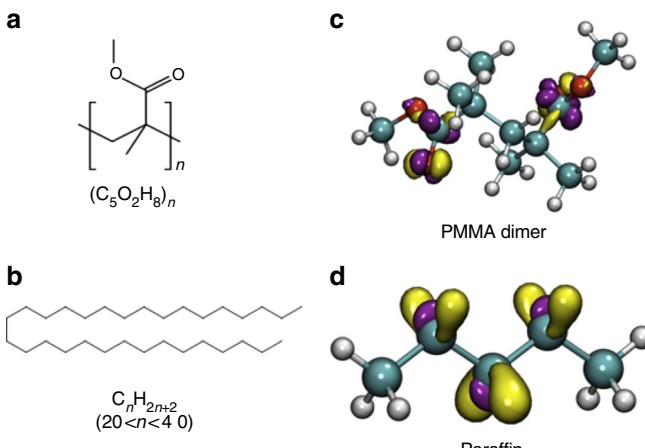

**Fig. 5** Chemical structures, formulae, and isosurfaces of a PMMA dimer and a paraffin molecule. Chemical structures and formulae of **a** PMMA and **b** paraffin, and the isosurfaces of the Fukui function for **c** a PMMA dimer and **d** a paraffin molecule. Carbon atoms in green, hydrogen atoms in gray, oxygen atoms in red. Yellow isosurfaces represent those regions of the molecule that undergo electrophilic attack upon chemical reaction, and purple represents regions that undergo nucleophilic attack

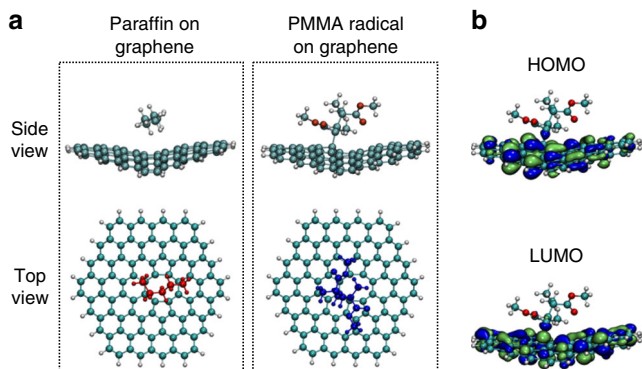

**Fig. 6** Adsorption ability of PMMA radical and paraffin molecule on graphene. **a** Adsorption of a paraffin molecule and PMMA radical on hexagonal graphene flakes with hydrogen-functionalized edges and a vacancy defect. Only the PMMA radical formed a covalent bond with the defective graphene flake, as verified by **b** the shared HOMO and LUMO of the bonded structure. Color code in the side view of **a**: carbon atoms in dark green, hydrogen atoms in gray, oxygen atoms in red. Color code in the top view of **a**: paraffin in red, PMMA radical in dark blue. Color code in **b**, the light green corresponds to regions of space where the phase of the wave function is positive, and the blue color corresponds to regions of space where the phase of the wave function is negative

**PMMA-supported graphene transfer**. First, a layer of polymethylmethacrylate (PMMA) 950 A5 (Microchem Inc.) was spun at 2500 rpm for 1 min on the graphene synthesized above Cu foil. The PMMA-coated graphene was then baked in an oven at 80 °C for 1 h. Next, the sample was floated on top of Cu etchant (Copper Etchant TFB, Transense) for 30 min to remove the growth substrate. The PMMA-supported graphene film was then rinsed with deionized water multiple times. Subsequently, the destination substrate (i.e. Si/SiO2 substrate) was contacted with the PMMA-supported graphene film and the sample was initially dried with nitrogen gun blowing, followed by an oven baking at 80 °C for at least 8 h. Subsequently, the sample was soaked in acetone at room temperature for 6 h to remove the PMMA support layer. The sample was then rinsed with IPA followed by nitrogen blow dry. We note that all process parameters chosen for the PMMA-supported graphene transfer in this work were adopted from earlier

reports[9,10,16,20,26]. A separate experiment was conducted to further verify the advantageous of paraffin support layers over PMMA (Supplementary Note 1).

**Fabrication of back-gated graphene field-effect transistors**. Graphene synthesized on Cu foil was first transferred on an oxidized degenerately p-doped silicon substrate with 285 nm thick $SiO_2$ using either paraffin or PMMA support layers. The sample was then spin-coated with a 300 nm thick layer of poly-methylmethacrylate (PMMA) 950 A5 (Microchem Inc.) and baked at 180 °C on a hotplate for 180 s. The source/drain contacts on graphene were delineated using electron beam lithography (EBL) and metallized with 20/30 nm of Ni/Au, followed by a 12-hour lift-off process in acetone at room temperature. Subsequently, the graphene film was delineated into a 5 μm wide ribbon using EBL followed by oxygen plasma etching. Finally, the sample was soaked in acetone for more than 12 h at room temperature to remove the PMMA electron beam resist. The dimensions of all graphene devices were kept constant in this work. The graphene channel and contact widths are 5 μm, while the channel length ranges from 1 to 11 μm. Ni was chosen as electrode as it is one of the metals that has been shown to provide reliable metal contacts to graphene[31]. No annealing is performed on any graphene devices prior to electrical measurements.

**Graphene characterization**. All Raman spectra and maps were taken using a WITec Alpha300R system. The laser excitation wavelength and grating density were 532 nm laser line and 600 mm$^{-1}$, respectively. The laser beam size was ~500 nm and the laser power on the sample was adjusted to around 0.75 mW to avoid laser-induced heating. Atomic force microscopy (AFM) topography image was obtained using Veeco Digital Instrument Nanoscope III in tapping mode.

**Differential scanning calorimeter**. The measurement was run at a constant heating rate of 10 °C min$^{-1}$ from 0 to 100 °C in a nitrogen gas atmosphere.

**DFT calculations**. DFT calculations were performed using the ORCA computational package[32]. The B3LYP functional was employed together with the 6-31 G* basis set to compute the optimized geometries, energies, and orbitals. All calculations were performed in implicit solvent using the conductor-like polarizable continuum (CPCM) model and the basis set superposition error (BSSE) was corrected with the geometrical counterpoise (gCP) method.

**Fukui function**. The dual descriptor of the Fukui function, a local conceptual DFT reactivity descriptor, was determined to spatially locate where eventual chemical reactions would take place. Originally introduced by Parr and Yang, the Fukui function can be interpreted in two ways: the sensitivity of the chemical potential of a system to an external perturbation at a particular point, or the change of the electron density at each point around the molecule when the total number of electrons is altered. For our purposes, the dual descriptor of the Fukui function detailed the different locations where each molecule would undergo nucleophilic or electrophilic attacks upon chemical reaction. The Fukui function was determined from electron density differences between the neutral and the positively- and negatively-charged PMMA or paraffin molecule, according to the following expression:

$$f(r) = f_+(r) - f_-(r) = [\rho_{N+1}(r) - \rho_N(r)] - [\rho_N(r) - \rho_{N-1}(r)]. \quad (3)$$

**Chemical hardness**. The chemical hardness, a global conceptual DFT descriptor, characterizes the polarizability of molecules as a measure of the resistance of a chemical species to changes in its electronic configuration. It can be interpreted as the global tendency of a molecule to give or accept electrons. Molecules with low polarizability are considered to be "hard" while molecules with high polarizability are conversely "soft". The chemical hardness is formally calculated from the ionization energy and the electron affinity. Based on Koopman's theorem, the first ionization energy can be approximated to the energy of the HOMO, while the electron affinity can be approximated to the energy of the LUMO. Thus, the chemical hardness was approximated from the energy "gap" between the HOMO and LUMO of the molecule,

$$\eta = \frac{1}{2}\left(U'_{HOMO} - U'_{LUMO}\right). \quad (4)$$

## Data availability

The data that support the findings of this study are available from the corresponding author upon request.

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

## Acknowledgements

J.K. acknowledges the support from AFOSR FATE MURI, Grant No. FA9550-15-1-0514, NSF DMR/ECCS–1509197, the Center for Energy Efficient Electronics Science (NSF Award 0939514), the King Abdullah University of Science and Technology (No. OSR-2015-CRG4-2634), and U.S. Army Research Office through the MIT Institute for Soldier Nanotechnologies (Grant No. 023674). J.-Y.H. acknowledges the support from Korea Research Institute of Chemical Technology (KRICT) project no. KK1801-G01 and the Basic Science Research Program through the National Research Foundation of Korea (NRF) funded by the Ministry of Education (NRF-2017R1C1B2007153). J.Y., F.J.M.-M, and M.J.B also acknowledge support from AFOSR FATE MURI, Grant No. FA9550-15-1-0514 and the US Department of Defense, Office of Naval Research (N00014–16–1–233). The work is partially performed at MIT Microsystems Technology Laboratories (MTL) and Center for Nanoscale Systems (CNS), Harvard University. Computational simulations were performed on the Extreme Science and Engineering Discovery Environment (XSEDE), which is supported by the National Science Foundation grant number ACI-1053575, the MIT Engaging Cluster, Singapore's A*STAR Computational Resource Centre, and Singapore's National Supercomputing Centre.

## Author contributions

W.S.L., H.W., J.-Y.H., and J.K. conceived the idea. W.S.L., H.W., A.Z., P.C.S., Y.M., T.P., J.-Y.H and J.K. performed the experiments and analyzed the data. J.Y., F.M.M. and M.J.B. performed the theoretical computations. W.S.L., H.W., J.Y., F.M.M. and J.K. wrote the manuscript. All authors contributed to the overall scientific interpretation and edited the manuscript.

## Additional information

**Competing interests:** The authors declare no competing interests.

