## [Peer Review File · Nature Communications]

Reviewers' comments:

Reviewer #1 (Remarks to the Author):

The authors describe an improvement to their previously published transfer technique 10.1002/adma.201505527 wherein the thermal expansion of an EVA polymer layer was used to reduce the amount of thermal-expansion-induced rippling in a CVD graphene layer. In this manuscript the EVA has been substituted for paraffin, and substantial device performance improvements over the author's PMMA transfers are shown, the differences being ascribed to the low reactivity of paraffin versus PMMA - a view supported by DFT calculations.

*** Although the presented technique might be promising and shows some apparently excellent results in terms of device properties I believe there are some serious issues with the performed experimental comparison and the description of the work. Overall I believe the similarity to previous work disqualifies this work for consideration here. ***

In addition, I find the following points which should be addressed before the manuscript is considered for publication elsewhere.

"heat reduces the surface tension of water" - in this case the expected reduction might be around 5%, going from room temp to 40°C, do the authors consider this significant?

"...electron mobility that is at least 4-fold higher than that of PMMA-transferred graphene..." - the authors should stress that the comparison is for the PMMA work performed here; literature values for μ_{FE} approaching or exceeding 7000 cm²/Vs have been reported for both e and h, e.g. 10.1038/ncomms1702 , 10.1038/s41598-017-18444-1 .

"In this work, CVD monolayer graphene grown on Cu foil was used [...] despite being polycrystalline and having inferior electrical properties" - with respect to what other kind(s) of graphene, exfoliated? SiC?

Since attempts are made to directly compare PMMA and paraffin transfers, I note a large number of differences between the treatment of the two substrates, some of which could be better controlled for in the experiments if we should compare apples with apples.

- The heat gun is only employed during the paraffin transfer - the substrate temperature is unknown (as are parameters which could be easily specified, such as the power of the heat gun, the distance from the substrate etc.)
- The paraffin transferred sample might benefit from the 1h soak in 40°C DI water which the PMMA sample does not get.
- The paraffin sample is baked at 40°C 24h, whereas the PMMA sample is baked at 80°C for 8h - why the difference, particularly given 80°C is below the T_g of PMMA 950K but 40°C is probably above the T_g for most paraffins?
- The paraffin layer is soaked in hexane for an unspecified period, and the PMMA layer in acetone for 6 hours. How are the typically highly contaminating acetone residues subsequently removed from the PMMA sample? Why not give both samples a hexane and acetone treatment to control for influences here?

Device fabrication steps also include application of PMMA as an e-beam resist, interactions of the device fabrication steps with the issues above should be addressed.

"Finally, the sample was soaked in warm acetone (60 °C) for more than 12 hours to remove the PMMA electron beam resist" Why was the same procedure not employed to remove PMMA residues from the PMMA transferred graphene if this is known to work well, rather than the stated 6h at room temperature?

Description of the production of van der Pauw type samples and their measurement (μH and R_s) is absent from the Methods section and details given in the main body text are inadequate to repeat the experiments. How was the magnetic field applied? How were the contacts applied? etc.

Critically, the material source and parameters of the paraffin used are not specified.

The written presentation of the DFT calculations in the main body text should be revised for brevity and clarity.

Reviewer #2 (Remarks to the Author):

The authors show in this body of work that CVD-grown graphene can be transferred from a copper substrate using a sacrificial paraffin support layer to produce graphene devices with improved electronic properties.

Although this work is interesting and thorough in terms of its electrical characterisation, there are areas that need significant address.

There are numerous instances of claims of less contamination of the graphene layer using this paraffin transfer method compared to a comparable PMMA transfer method, however, at no point is there any measurement of the chemical properties of the surface. For these claims to be directly evidenced, a technique such as X-ray photoelectron spectroscopy (XPS) should be employed to compare the chemical species present for each transfer process. Even better would be a technique such as time-of-flight secondary ion mass spectrometry (ToF-SIMS) to determine what molecular species are left on the surface of the graphene after transfer and removal and whether there is paraffin still present.

This chemical assessment would allow the authors to better understand whether the improved electronic properties are actually just due to the reduction in defects (wrinkles) in graphene. AFM is used as the primary method to determine the contamination on the surface, but depending on the AFM measurement parameters used, surfaces with similar topography could seem very different in the resulting AFM images. In fact, there is essentially no description of the AFM measurements in the manuscript, not even if the measurements are made in 'tapping-mode' or 'contact mode'. Contact mode is known to move contamination from the imaged areas of surface, but not necessarily have the same effect on graphene wrinkles. There are also no AFM profiles shown throughout the manuscript and so the features observed in the image are harder to understand. The term 'nano-sized' wrinkles should also be defined, particularly if the 'nano' refers to the length or width of the wrinkle.

Furthermore, some features in the AFM measurements are attributed to being bilayer islands of graphene (Figure 4b). Although possible, there are no height measurements shown to confirm this. In fact, they could be instead islands of contamination, as alkanes have been known to form self-assembled monolayers on a graphitic surface for many years (e.g. Rabe & Buchholz, *Science* (1991), 253, 424-427, DOI: 10.1126/science.253.5018.424). Figure 4a does not appear to show the same islands, even though the produced graphene should be the same. This is another issue that could be solved using a direct chemical analysis method to characterise the surface.

Another important area that needs addressing is the title and the novelty of this paper. Paraffin-enabled transfer of CVD-grown graphene has been performed before, in fact published by the authors themselves (Martins et al., *Proc Natl Acad Sci U S A.* (2013) 110, 17762–17767, DOI: 10.1073/pnas.1306508110), but yet, not referenced in this body of work. The novelty of this work is more the reduction in wrinkles of the transferred graphene, which should be reflected in the title.

Further work is also needed in the attention to the treatment of the electrical characterisation results. The points shown in Figure 2 do not have error bars to provide an idea of the uncertainty in the measurement itself to show if the variations are significant. Interestingly there is a comment 'The sheet resistance of paraffin-transferred graphene was considerably lower with a narrower distribution in comparison with that of PMMA-transferred graphene ($502 \pm 12 \Omega/\square$ versus $533 \pm 34 \Omega/\square$).'. Although this is indeed a narrower distribution for the paraffin-transferred graphene, the two methods essentially show the same final value of sheet resistance, as they are both within error of each other, so the 'considerably lower' sheet resistance claim should be removed for the data in Figure 2b. Figure 2c and 2d would perhaps back this claim, but an average value and standard deviation is not provided for the PMMA transfer, only a range.

Two more minor comments are on (1) the English used, which at times seems to need a more thorough proof-read (example 'the sample was transferred onto a 40 °C deionized water' on page 3) and (2) improved photos in Figure 1c, as they appear to be of a transparent petri dish with out-of-focus items on a table on the other side of the petri dish, making it hard to see the main features of the photos themselves.

Reviewer #3 (Remarks to the Author):

This is a very nice paper that presents an important advance in graphene processing technology. The paper is elegant in its simplicity. Using the high thermal expansion of a paraffin handle to stretch graphene after Cu etching effectively removes wrinkles that would otherwise be present after graphene growth. Though the stretching due to thermal expansion of the paraffin is not sufficient to tear the graphene, temperatures lower than 40 degC could be used if that were the case. A quick online search shows that paraffin has a melting point that ranges from 39-68 degC. The authors should state the melting point of the paraffin used in their study.

The authors provide compelling evidence for the efficacy of using paraffin. The data in figure 2 show much higher hall mobilities and lower carrier concentrations and lower and more uniform sheet resistances. (Minor point: the main text states the data in fig. 2 was obtained from 10 samples, however I see only 6 data points for PMMA and paraffin in figs 2a,b.)

The backgated FET results in fig. 3 are also very nice, showing a much smaller and more tightly distributed Dirac voltage for the paraffin case, as well as much higher field effect mobility. The AFM images in figure 4 show direct physical evidence for the smoothing effect of the paraffin and the low strain and doping of the transferred graphene. Interestingly, small regions of bilayer graphene are seen in figure 4b, as mentioned by the authors.

The simulation part of the paper, presented in figures 5 and 6 is quite plausible and agrees nicely with the clean transfer using paraffin while explaining why PMMA transfers are prone to residue.

I believe this paper will be of broad interest not only to graphene researchers, but to groups working on wider ranges of 2D materials. I recommend publication as is with the addition of the paraffin melting temperature and clarification of the number of data points in figure 2.

Response to Reviewers' Comments (NCOMMS-18-29295)

We sincerely thank the reviewers for carefully reviewing our work. In the following, we provide new analyses and new data to address the concerns of the reviewers. The comments of the three reviewers are reproduced below, together with the authors' point-by-point response on changes made in the revised manuscript to address each reviewer's comments. We feel the manuscript is much improved now and hope our efforts can help to remove the doubts of the reviewers. The changes made in the manuscript and supplemental information are marked in blue font.

Reviewer #1 (Remarks to the Author):

The authors describe an improvement to their previously published transfer technique 10.1002/adma.201505527 wherein the thermal expansion of an EVA polymer layer was used to reduce the amount of thermal-expansion-induced rippling in a CVD graphene layer. In this manuscript the EVA has been substituted for paraffin, and substantial device performance improvements over the author's PMMA transfers are shown, the differences being ascribed to the low reactivity of paraffin versus PMMA - a view supported by DFT calculations.

Although the presented technique might be promising and shows some apparently excellent results in terms of device properties, I believe there are some serious issues with the performed experimental comparison and the description of the work. Overall I believe the similarity to previous work disqualifies this work for consideration here.

In addition, I find the following points which should be addressed before the manuscript is considered for publication elsewhere.

Response: We thank the reviewer for the comment here, we agree that there is a similarity between this work and our previous work (*Adv. Mater.* **28**, 2382-2392 (2015)) regarding the wrinkle removal in graphene. However, our work on the EVA-assisted transfer was tailored for transferring graphene onto rough surfaces, while in this work, we have found the homogeneous and enhanced electrical properties when using paraffin as a support layer for transfers. This represents an important finding for scalable production of 2D materials for the fabrication of highly-reliable ubiquitous electronics, which is of broad interest of the scientific community and hence the work is well suited for publication in *Nature Communications*.

We have also considered all the issues that the reviewer listed and realized that we did not include sufficient experimental details / work descriptions in our earlier submission, leading to some misunderstandings. We sincerely appreciate the reviewer's efforts for the careful review of our work. Based on the questions, we have improved on the clarity of our work. We deeply appreciate the reviewer's comments which have significantly helped to improve our work.

"heat reduces the surface tension of water" - in this case the expected reduction might be around 5%, going from room temp to 40°C, do the authors consider this significant?

Response: We agree with the reviewer that it is not significant, and we thank the reviewer for this comment. We have thus deleted the sentence, *"In addition, heat reduces the surface tension of water, thereby minimizing the amount of water trapped between the graphene and the destination substrate.²⁵"* on manuscript page 2.

"...electron mobility that is at least 4-fold higher than that of PMMA-transferred graphene..." - the authors should stress that the comparison is for the PMMA work performed here; literature values for μ FE approaching or exceeding 7000 cm²/Vs have been reported for both e and h, e.g. 10.1038/ncomms1702, 10.1038/s41598-017-18444-1.

Response: We thank the reviewer for the suggestion and we have revised the sentence accordingly on manuscript page 2:

"...the electron mobility that is ~4-fold higher than that of our PMMA-transferred graphene..."

"In this work, CVD monolayer graphene grown on Cu foil was used [...] despite being polycrystalline and having inferior electrical properties" - with respect to what other kind(s) of graphene, exfoliated? SiC?

Response: We thank the reviewer for this comment, we meant the polycrystalline graphene has inferior properties to the single crystalline graphene. We have thus modified this sentence below to avoid confusion:

"In this work, CVD monolayer graphene grown on Cu foil was used because graphene synthesized in this manner is the most widely used source of large-area graphene in state-of-the-art research and industrial development, despite being polycrystalline and having modest electrical properties."

Since attempts are made to directly compare PMMA and paraffin transfers, I note a large number of differences between the treatment of the two substrates, some of which could be better controlled for in the experiments if we should compare apples with apples.

Response: We thank the reviewer for this comment, and in the following we have made changes to address this concern.

- The heat gun is only employed during the paraffin transfer - the substrate temperature is unknown (as are parameters which could be easily specified, such as the power of the heat gun, the distance from the substrate etc.)

Response: The substrate temperature was measured to be ~ 50 °C and we have added this note in the Methods on manuscript page 16:

"Heat gun was used to blow on the sample making the paraffin melts again and the substrate temperature was measured to be ~ 50 °C."

- The paraffin transferred sample might benefit from the 1h soak in 40°C DI water which the PMMA sample does not get.

Response: We have the same inference as the reviewer earlier on. Indeed, we have attempted to transfer PMMA/graphene sample onto 40 °C deionized water surface and kept for the same duration. However, there is no observable difference with the controlled PMMA/graphene sample transferred at room temperature.

- The paraffin sample is baked at 40°C 24h, whereas the PMMA sample is baked at 80°C for 8h - why the difference, particularly given 80°C is below the T_g of PMMA 950k but 40°C is probably above the T_g for most paraffins?

Response: This step of baking aims to slowly dry the water trapped at the interface of graphene and the destination substrate, to minimize the cracks/tears generation in graphene. Specifically, 40 °C was chosen for the paraffin/graphene samples because the melting point of paraffin used throughout the study is about 45 °C, extracted from a differential scanning calorimeter study (new Supplemental Fig. 4). Temperatures higher than 40 °C may cause the paraffin support layer melts, while that of lower may prolong the baking process. On the other hand, 80 °C was chosen for the PMMA/graphene samples as temperatures higher than 80 °C evaporate water at a very fast rate, resulting in severe cracks/tears in graphene.

Supplemental Fig. 4 | Differential scanning calorimetry (DSC) measurement of our paraffin sample.

- The paraffin layer is soaked in hexane for an unspecified period, and the PMMA layer in acetone for 6 hours. How are the typically highly contaminating acetone residues subsequently removed from the PMMA sample? Why not give both samples a hexane and acetone treatment to control for influences here?

Response: The paraffin/graphene samples were soaked in hexane for 2 h and the acetone residues were removed by rinsing the samples with IPA. We have added two notes in the Methods on manuscript page 16:

1. *“Finally, the paraffin support layer was removed using hexane solution for 2 h, leaving monolayer graphene on the destination substrate.”*
2. *“Subsequently, the sample was soaked in acetone at room temperature for 6 h to remove the PMMA support layer. The sample was then rinsed with IPA followed by nitrogen blow dry.”*

Furthermore, acetone cannot dissolve paraffin and hexane cannot dissolve PMMA. We were unable to find a solvent that can dissolve both paraffin and PMMA. Therefore, different solvents were used for the paraffin- and PMMA-supported transfers.

Device fabrication steps also include application of PMMA as an e-beam resist, interactions of the device fabrication steps with the issues above should be addressed.

Response: We agree with the reviewer that zero PMMA interaction is important for graphene. Remarkably, field-effect transistors fabricated on our paraffin-transferred graphene exhibits enhanced electrical performance, despite being contaminated with PMMA once during the device fabrication processes. Also, on manuscript page 4, we have discussed about this:

“The electron mobility of paraffin-transferred graphene was 4 times higher than that of PMMA, regardless of the device channel length. It is worth noting that all FETs fabricated on paraffin-transferred graphene in this study were contaminated with PMMA once during the device fabrication processes.”

"Finally, the sample was soaked in warm acetone (60 °C) for more than 12 hours to remove the PMMA electron beam resist" Why was the same procedure not employed to remove PMMA residues from the PMMA transferred graphene if this is known to work well, rather than the stated 6h at room temperature?

Response: We thank the reviewer for this comment. When we wrote the manuscript, the experimental parts were adopted from the authors' work in previous lab (*Nano Lett.* **14**, 3840-3847 (2014): warm acetone (60 °C)) and we forgot to change this particular details to reflect the experimental conditions in our lab now (*J. Am. Chem. Soc.* **14**, 12354–12358 (2018): acetone at room temperature; we have compared the results and did not find significant improvement in terms of both polymer residues and device performance on graphene. Thereafter, we have chose to perform acetone removal process at room temperature). We sincerely apologize for this careless mistake. We have thus corrected the experimental details on the manuscript page 17:

“Finally, the sample was soaked in acetone for more than 12 h at room temperature to remove the PMMA electron beam resist.”

Description of the production of van der Pauw type samples and their measurement (μH and R_s) is absent from the Methods section and details given in the main body text are inadequate to repeat the experiments. How was the magnetic field applied? How were the contacts applied? etc.

Response: We thank the reviewer to bringing up this point and we have added the detailed description of Hall measurements in the manuscript page 3:

“Four small contact regions were then made by pressing a thin slice of indium wire with metal tweezers at the corners of each squared graphene film (contact length is less than 7% of the side length of the graphene film). Subsequently, resistivity and Hall measurements were made on each paraffin-transferred graphene film in the presence of 2,800 Gauss magnetic field to extract its sheet resistance, carrier concentration, and Hall mobility values. During the Hall measurements, the magnetic field is applied by attaching a piece of magnet underneath the sample with desired polarity.”

Critically, the material source and parameters of the paraffin used are not specified.

Response: We have added the required information in the Methods on page 16:

“To prepare the paraffin solution, paraffin pastilles were purchased from Sigma-Aldrich 18634 and kept in an 80 °C oven.”

The written presentation of the DFT calculations in the main body text should be revised for brevity and clarity.

Response: Following the reviewer’s suggestion, we have moved some of the details to Supplemental Information and revised the paragraph on DFT calculations for brevity and clarity on manuscript page 6:

“To unveil the fundamental chemical mechanisms underlying the distinct contamination level in PMMA- and paraffin-transferred graphene, we performed DFT calculations to optimize the geometries and to determine conceptual DFT chemical reactivity descriptors for both PMMA and paraffin chemical structures (Fig. 5a, b). Thus, the dual descriptor of the Fukui function and the chemical hardness were calculated for PMMA dimers and a short chain of paraffin. From Fig. 5c-d, the dual descriptor of the Fukui function for the PMMA dimer indicated that the carbonyl groups were very reactive compared to the rest of the molecule, whereas the reactivity of paraffin was generally equivalent throughout the molecule. These concentrations of local reactivity in the carbonyl functional groups increases the ability of PMMA to undergo chemical reactions in that part of the molecule. Furthermore, the chemical hardness for the PMMA dimer was 88.07 kcal/mol, in contrast to the much higher chemical hardness of 157.04 kcal/mol for paraffin. This indicates a much higher resistance of paraffin to get involved in chemical reactions that imply any transfer of electrons. Taken together, these results on the local and global chemical reactivity demonstrated that PMMA was significantly more likely to react in the presence of any eventual chemically reactive species. Our DFT calculations further indicate that the adsorption energies indicated that PMMA adsorbed more strongly to the surface of graphene with an adsorption energy that was 1.56 kcal/mol higher than that of paraffin. As shown in Fig. 6a, formation of a paraffin-graphene interface is mostly driven by non-covalent interactions and no covalent bonds would be formed between paraffin and graphene, even in the

presence of a vacancy in graphene. In contrast, PMMA radicals bond covalently to the defective graphene, even if the carbon atoms surrounding the vacancy defect rearranged themselves to form more stable 5- and 8-membered rings (Fig. 6a), as verified by the shared highest occupied molecular orbitals (HOMO) and lowest unoccupied molecular orbitals (LUMO) (Fig. 6b). All these calculations support the idea that the appearance of large amounts of PMMA contaminants on the PMMA-transferred graphene was likely due to the combined effects of stronger non-covalent interaction between PMMA and graphene, higher reactivity of PMMA, and possible covalent bonding between PMMA radicals and the vacancy defects in graphene at elevated temperatures. In sharp contrast, paraffin's chemical properties of lower reactivity, lower non-covalent affinity to graphene, and lesser likelihood of forming stable radicals that lead to covalent bonding, were remarkably beneficial for producing cleaner, large-area graphene sheets.”

Again, we sincerely thank the reviewer for bringing up all these important questions, which helps to improve the robustness of the reported work.

Reviewer #2 (Remarks to the Author):

The authors show in this body of work that CVD-grown graphene can be transferred from a copper substrate using a sacrificial paraffin support layer to produce graphene devices with improved electronic properties. Although this work is interesting and thorough in terms of its electrical characterisation, there are areas that need significant address.

There are numerous instances of claims of less contamination of the graphene layer using this paraffin transfer method compared to a comparable PMMA transfer method, however, at no point is there any measurement of the chemical properties of the surface. For these claims to be directly evidenced, a technique such as X-ray photoelectron spectroscopy (XPS) should be employed to compare the chemical species present for each transfer process. Even better would be a technique such as time-of-flight secondary ion mass spectrometry (ToF-SIMS) to determine what molecular species are left on the surface of the graphene after transfer and removal and whether there is paraffin still present.

Response: We thank the reviewer for this comment and agree that using direct chemical analysis method to characterise the surface of transferred graphene is important. Following the reviewer's suggestion, we have conducted XPS studies. The results are plotted in Supplemental Fig. 1 and discussed in the manuscript page 5:

"... In Supplemental Fig. 1, we plotted the X-ray photoelectron spectroscopy (XPS) spectrum of PMMA- and paraffin-supported graphene transferred on Si/SiO₂ substrate. As can be seen, our paraffin-transferred graphene sample has no C=O and O=C-OH peaks and the C-OH peak intensity is relatively weak, indicating lower levels of polymer contamination."

Supplemental Fig. 1 | X-ray photoelectron spectroscopy (XPS) spectrum of **a**, PMMA- and **b**, paraffin-supported graphene transferred on Si/SiO₂ substrate. Arrows in **a** indicate the peaks related to PMMA residual.

This chemical assessment would allow the authors to better understand whether the improved electronic properties are actually just due to the reduction in defects (wrinkles) in graphene. AFM is used as the primary method to determine the contamination on the surface, but depending on the AFM measurement parameters used, surfaces with similar topography could seem very different in the resulting AFM images. In fact, there is essentially no description of the AFM measurements in the manuscript, not even if the measurements are made in ‘tapping-mode’ or ‘contact mode’. Contact mode is known to move contamination from the imaged areas of surface, but not necessarily have the same effect on graphene wrinkles. There are also no AFM profiles shown throughout the manuscript and so the features observed in the image are harder to understand. The term ‘nano-sized’ wrinkles should also be defined, particularly if the ‘nano’ refers to the length or width of the wrinkle.

Response: We appreciate the very helpful suggestions from the reviewer. We agree with the reviewer that these are important and we have made the modifications accordingly. First, we have added the description on tapping mode AFM in the Methods on manuscript page 17:

“Atomic force microscopy (AFM) topography image was obtained using Veeco Digital Instrument Nanoscope III in tapping mode.”

Second, we have included the AFM height profiles in Supplemental Fig. 2 (new):

Supplemental Fig. 2 | a, Typical AFM topography image of a paraffin-transferred graphene on a Si/SiO₂ substrate. **b**, Height profile along the line marked. **c**, Raman spectra of the two spots marked. As can be seen, our paraffin-transferred graphene has a clean and smooth surface such that 1-μm-sized bilayer graphene domains can be clearly observed under AFM and confirmed by Raman analysis.

Third, we agree with the reviewer that “nano-sized” is not an accurate term, we have thus revised the text on manuscript page 5 as copied below:

“In Fig. 4b, over a graphene area of 20 × 20 μm², there were very few wrinkles measuring more than 10 μm in length. More surprisingly, smaller wrinkles in size (both height and width) are not observable on the graphene surface. For the removal of graphene wrinkles, we inferred that the paraffin support layer could effectively smoothen these wrinkles through thermal expansion and releasing the compressive strain generated in graphene during the cooling step in the CVD growth process.”

Furthermore, some features in the AFM measurements are attributed to being bilayer islands of graphene (Figure 4b). Although possible, there are no height measurements shown to confirm this. In fact, they could be instead islands of contamination, as alkanes have been known to form self-assembled monolayers on a graphitic surface for many years (e.g. Rabe & Buchholz, Science (1991), 253, 424-427, DOI: 10.1126/science.253.5018.424). Figure 4a does not appear to show the same islands, even though the produced graphene should be the same. This is another issue that could be solved using a direct chemical analysis method to characterise the surface.

Response: We thank the reviewer for bringing up this point. We agree with the reviewer that the distribution of bilayer graphene domains are different in the original Fig. 4a, b. As can be seen in Supplemental Fig. 6c, d (reprinted below), the bilayer graphene domains are randomly distributed in a CVD-grown graphene transferred onto the same piece of $1 \times 1 \text{ cm}^2$ Si/SiO₂ substrate. Hence, it is very challenging to obtain two AFM images of graphene with the same distribution of bilayer graphene domains for comparisons, even if they were obtained from the same piece of transferred graphene. To avoid confusion, we have made the following modifications:

1. Modified the sentence relevant to bilayer graphene domains on manuscript page 5:

Original sentence: *“Compared to PMMA-transferred graphene, paraffin-transferred graphene had a much cleaner surface such that 1- μm -sized bilayer graphene domains were clearly observed under AFM, even in the absence of annealing (Figure 4b).”*

Revised sentence: *“Compared to PMMA-transferred graphene, paraffin-transferred graphene had a much cleaner surface (Fig. 4b and Supplemental Fig. 2).”*

2. Moved the original Fig. 4b to Supplemental Fig. 2 and added Raman analysis to confirm those islands are bilayer graphene domains, not alkane.
3. Added a new Fig. 4b showing a paraffin-transferred graphene region with no bilayer graphene islands (reprinted below).

Supplemental Fig. 6 | Comparison of PMMA- and paraffin-transferred graphene. **a-b**, Typical optical images of PMMA-transferred graphene on a Si/SiO₂ substrate. **c-d**, Typical optical images of paraffin-transferred graphene on Si/SiO₂ substrate. Blue arrows indicate some of the bilayer graphene (BLG) domains. Red arrows indicate the presence of polymer residues. Dotted circles indicate minor cracks randomly generated in the graphene due to the incomplete removal of residual water.

Fig. 4 | Materials characterization of the PMMA- and paraffin-transferred graphene on Si/SiO₂ substrate. **a-b**, Typical AFM height profile images of graphene film transferred with **a**, PMMA and **b**, paraffin support layers. **c-d**, Correlation map of the Raman G and 2D peak positions of graphene transferred with **c**, PMMA and **d**, paraffin support layers. A total of 3600 Raman spectra were taken from each type of transferred graphene and the corresponding G peak position (ω_G), 2D peak position (ω_{2D}), and 2D peak's full width at half maximum (Γ_{2D}) were extracted. The black circle represents the G and 2D peak positions of an intrinsic graphene, where graphene has neither doping nor strain.

Another important area that needs addressing is the title and the novelty of this paper. Paraffin-enabled transfer of CVD-grown graphene has been performed before, in fact published by the authors themselves (Martins et al., Proc Natl Acad Sci U S A. (2013) 110, 17762–17767, DOI: 10.1073/pnas.1306508110), but yet, not referenced in this body of work. The novelty of this work is more the reduction in wrinkles of the transferred graphene, which should be reflected in the title.

Response: We apologize for the confusion here. Our previous work (*Proc. Natl. Acad. Sci. U.S.A.* **110**, 17762–17767 (2013)) was developed to transfer graphene directly onto a variety of flexible substrates, among them paraffin was one of the substrates. However, in that work, paraffin was used only as a substrate. In contrast, in our new paraffin-enabled graphene transfer technique, the paraffin is used as a sacrificial layer, instead of being the substrate. To avoid confusion, we have added a note in the Introduction of the manuscript:

“Here, we report a paraffin-enabled transfer technique that concurrently addresses both support layer contamination and wrinkling in graphene, enabling transfers of large-area graphene with homogeneous and enhanced electrical properties. We propose to use paraffin as a transfer support layer based on two rationales: (1) paraffin is an alkane with a simple unreactive chemical structure and (2) it has high thermal expansion coefficient. We have shown earlier that paraffin can be used as a flexible substrate to transfer graphene onto it,²² and in this work, paraffin is used as a sacrificial transfer support layer.”

Further work is also needed in the attention to the treatment of the electrical characterisation results. The points shown in Figure 2 do not have error bars to provide an idea of the uncertainty in the measurement itself to show if the variations are significant. Interestingly there is a comment ‘The sheet resistance of paraffin-transferred graphene was considerably lower with a narrower distribution in comparison with that of PMMA-transferred graphene ($502 \pm 12 \Omega/\square$ versus $533 \pm 34 \Omega/\square$).’. Although this is indeed a narrower distribution for the paraffin-transferred graphene, the two methods essentially show the same final value of sheet resistance, as they are both within error of each other, so the ‘considerably lower’ sheet resistance claim should be removed for the data in Figure 2b. Figure 2c and 2d would perhaps back this claim, but an average value and standard deviation is not provided for the PMMA transfer, only a range.

Response: We agree with the reviewer that the comparison of $502 \pm 12 \Omega/\square$ and $533 \pm 34 \Omega/\square$ is very misleading. In fact, these two average and standard deviation values are not reliable as they were calculated from only 6 data points each (Fig. 2b). Therefore, average and standard deviation values obtained from 800 sheet resistance values each shown in Fig. 2c, d are more reliable to support our claim on “considerably lower”. We have thus revised the paragraph on manuscript page 4 as follow:

“For the same group of graphene samples, sheet resistance was also extracted using the van der Pauw method and plotted in Fig. 2b. The sheet resistance of paraffin-transferred graphene was considerably lower with a narrower distribution in comparison with that of PMMA-transferred graphene. To gain further insights into the spatial distribution of sheet resistances across these large-area graphene films transferred with different support layers, 800 sheet resistance values were measured using a four-point probe tool over graphene films with dimensions of $40 \times 20 \text{ mm}^2$, with 1 mm step size in both the x- and y-directions. The spatial sheet resistance maps of both PMMA- and paraffin-transferred graphene are plotted in Fig. 2c,d, respectively, with the same color intensity bar. For PMMA-transferred graphene, the measured sheet

resistances varied broadly from 446 to 916 Ω/\square (average = $600 \pm 95 \Omega/\square$), with random areas showing relatively high average sheet resistances that could be due to damage on graphene induced by the transfer process, including minor cracks, wrinkles and clustering of PMMA residues. In contrast, the paraffin-transferred graphene film exhibited lower sheet resistances with a much narrower deviation (average = $507 \pm 17 \Omega/\square$) that were distributed homogeneously for the same sample size, with slightly increased sheet resistances along the edges of the graphene film.”

Two more minor comments are on (1) the English used, which at times seems to need a more thorough proof-read (example ‘the sample was transferred onto a 40 °C deionized water’ on page 3) and (2) improved photos in Figure 1c, as they appear to be of a transparent petri dish with out-of-focus items on a table on the other side of the petri dish, making it hard to see the main features of the photos themselves.

Response: We thank the reviewer for pointing out these issues. We have conducted a more thorough English proof-reading. The grammar mistakes on manuscript page 3 and photos in Fig. 1c have been corrected accordingly:

“Subsequently, the sample was transferred onto 40 °C deionized water and kept at the same temperature for 1 h, where the paraffin layer remained at a solid state (Fig. 1c).”

Fig. 1 | Our paraffin-enabled graphene transfer technology. **a**, Schematics showing the process of paraffin-assisted graphene transfer. **b**, Schematics showing the effect of paraffin’s thermal expansion on graphene wrinkle. **c**, Photographs of a typical paraffin-supported graphene film floated on water at different temperatures as indicated confirming that the paraffin layer is still in solid state at ~40 °C.

Reviewer #3 (Remarks to the Author):

This is a very nice paper that presents an important advance in graphene processing technology. The paper is elegant in its simplicity. Using the high thermal expansion of a paraffin handle to stretch graphene after Cu etching effectively removes wrinkles that would otherwise be present after graphene growth. Though the stretching due to thermal expansion of the paraffin is not sufficient to tear the graphene, temperatures lower than 40 degC could be used if that were the case. A quick online search shows that paraffin has a melting point that ranges from 39-68 degC. The authors should state the melting point of the paraffin used in their study.

Response: We appreciate the reviewer's comment to help us improving our manuscript. We have conducted differential scanning calorimetry experiments on our paraffin sample and the result is plotted in Supplemental Fig. 4. As can be seen, the melting point of paraffin used in our work is ~ 45 °C.

Supplemental Fig. 4 | Differential scanning calorimetry (DSC) measurement of our paraffin sample.

The authors provide compelling evidence for the efficacy of using paraffin. The data in figure 2 show much higher hall mobilities and lower carrier concentrations and lower and more uniform sheet resistances. (Minor point: the main text states the data in fig. 2 was obtained from 10 samples, however I see only 6 data points for PMMA and paraffin in figs 2a,b.)

Response: We thank the reviewer for pointing out this mistake and which has been corrected in the manuscript page 3. The total samples measured was indeed 12 (6 for paraffin and 6 for PMMA), not 10.

The backgated FET results in fig. 3 are also very nice, showing a much smaller and more tightly distributed Dirac voltage for the paraffin case, as well as much higher field effect mobility. The AFM images in figure 4 show direct physical evidence for the smoothing effect of the paraffin and the low strain and doping of the transferred graphene. Interestingly, small regions of bilayer graphene are seen in figure 4b, as mentioned by the authors. The simulation part of the paper, presented in figures 5 and 6 is quite plausible and agrees nicely with the clean transfer using paraffin while explaining why PMMA transfers are prone to residue. I believe this paper will be of broad interest not only to graphene researchers, but to groups working on wider ranges of 2D materials. I recommend publication as is with the addition of the paraffin melting temperature and clarification of the number of data points in figure 2.

Response: We thank the reviewer for the very positive comment of this work.

Reviewers' comments:

Reviewer #1 (Remarks to the Author):

I would like to thank the authors for addressing my concerns, and am happy to be contradicted that the manuscript is suitable for publication in Nature Communications. I have a number of responses to the authors which I would like taken into consideration still.

re: "surface tension of water" - thank you for removing this sentence.

re: "4 fold higher mobility" - the new formulation is clearer.

re: "inferior/modest properties" - my point was really to ask what was being compared in the comparison: I understand now that polycrystalline graphene is being compared to monocrystalline graphene from the authors response (as opposed to CVD vs other sources), but I am not quite satisfied that this message comes across in the revised version.

re: "heat gun temperature" - not to be too pedantic, but this edit raises the question of how the substrate temperature was measured of course... was it just inferred from the melting paraffin? does it change during spinning?

re: "DI water soak" - this is unsatisfactory in my opinion: the authors state that the results are the same for PMMA for DI soak vs. no DI soak yet choose to present the results for the sample that is treated differently to the paraffin, i.e. deliberately compare apples and oranges?

re: "Tg" - Thank you for the new data showing the Tg of the paraffin used here. However, I feel that the point stands that to treat the samples fairly one would dry both PMMA and paraffin at 40°C for 24 hours - especially knowing (as stated in the response) that high temperatures can cause actual damage to the graphene!

re: "hexane vs acetone" - A 50/50 mix of hexane and acetone would dissolve the paraffin and the PMMA over 6 hours and provide the fairest test in my opinion, and both samples could be washed in IPA since all three solvents are freely miscible. The tendency of acetone to be dirty from dissolved plasticisers, self-aldolisation and dissolved water makes me worry that the current hexane-only sample is getting unfair treatment by avoiding acetone despite careful washing.

re: "PMMA contact during ebeam" - thank you for the changes - there is some evidence that PMMA residues might be relatively benign though, e.g. <https://doi.org/10.1088/2053-1583/1/3/035005>.

re: "removing ebeam resist at 60C for 12h" - still, despite the changes, readers might question that the ebeam resist layer is given 12 hours to dissolve, and a somewhat thicker PMMA support layer (judging from the spin speed) only given 6 hours.

re: "van der Pauw details" - thank you for the description of the method.

re: "paraffin source" - thank you

re: "DFT clarity" - moving this paragraph to the SI has improved readability

T. Booth

Reviewer #2 (Remarks to the Author):

I thank the authors for their changes to the manuscript, which are very thorough. I believe this work can now be published in Nature Communications.

Response to Reviewers' Comments (NCOMMS-18-29295A)

We sincerely thank the reviewers for carefully reviewing our work. In the following, we provide new data to address the concerns of the reviewers. The comments of the two reviewers are reproduced below, together with the authors' point-by-point response on changes made in the revised manuscript to address each reviewer's comments. We feel the manuscript is much improved now and hope it is now suitable for publication in *Nature Communications*. The changes made in the manuscript and supplemental information are marked in blue font.

Reviewer #1 (Remarks to the Author):

I would like to thank the authors for addressing my concerns, and am happy to be contradicted that the manuscript is suitable for publication in *Nature Communications*. I have a number of responses to the authors which I would like taken into consideration still.

Response: We are glad that the reviewer is now supportive of our work. We sincerely thank the reviewer for all his fruitful suggestions and comments, and in the following we have made changes to address the reviewer's concerns.

re: "surface tension of water" - thank you for removing this sentence.

re: "4 fold higher mobility" - the new formulation is clearer.

re:"inferior/modest properties" - my point was really to ask what was being compared in the comparison: I understand now that polycrystalline graphene is being compared to monocrystalline graphene from the authors response (as opposed to CVD vs other sources), but I am not quite satisfied that this message comes across in the revised version.

Response: We thank the reviewer for this comment. We have thus modified the sentence on manuscript page 2 to be more specific, it is copied below:

"In this work, CVD monolayer graphene grown on Cu foil was used because graphene synthesized in this manner is the most widely used source of large-area graphene in state-of-the-art research and industrial development, despite being polycrystalline and having lower mobility values."

re:"heat gun temperature" - not to be too pedantic, but this edit raises the question of how the substrate temperature was measured of course... was it just inferred from the melting paraffin? does it change during spinning?

Response: We thank the reviewer for this comment. On a separate testing, we attached a thermocouple to the substrate and blow it with heat gun for 2 min. We measured that the substrate temperature is constantly ~ 50 °C under the heat gun blow for 2 min. However, it is too challenging to measure the substrate temperature during spinning. We have thus modified the experimental details on the manuscript page 16:

“Heat gun was used to blow on the sample making the paraffin melts again and the substrate temperature was measured to be ~ 50 °C using a thermocouple. Once the paraffin on sample turns into liquid form, the sample was spun at 1,000 rpm for 2 min, with continuous heat gun blowing.”

re: "DI water soak" - this is unsatisfactory in my opinion: the authors state that the results are the same for PMMA for DI soak vs. no DI soak yet choose to present the results for the sample that is treated differently to the paraffin, i.e. deliberately compare apples and oranges?

Response: We thank the reviewer for this comment. We have thus added a note stating the rationale of parameters chosen for the PMMA-assisted graphene transfer in the Methods on manuscript page 17. Following the reviewer's suggestion, we have also conducted a new control experiment, namely "Modified PMMA-supported graphene transfer process", which has minimal process variants compared to the paraffin-enabled graphene transfer technique. In this control experiment, the PMMA-supported graphene also experienced the DI water soak. Specifically, the PMMA-supported graphene sample was transferred onto 40 °C deionized water surface and kept at the same temperature for at least 1 h, similar to that of the paraffin-transferred graphene. The results are detailed in Supplementary Note 1 and are also copied below. As can be seen in Supplemental Fig. 7, the graphene sample transferred with modified PMMA-assisted approach does not benefit from the "DI water soak" process, where polymer contamination and wrinkle can be clearly seen with a density similar to that of the conventional PMMA-transferred graphene, which was used in the manuscript. We hope this can help to address the reviewer's concern.

Manuscript page 17:

“PMMA-supported graphene transfer. First, a layer of polymethylmethacrylate (PMMA) 950 A5 (Microchem Inc.) was spun at 2,500 rpm for 1 min on the graphene synthesized above Cu foil. The PMMA-coated graphene was then baked in an oven at 80 °C for 1 h. Next, the sample was floated on top of Cu etchant (Copper Etchant TFB, Transense) for 30 min to remove the growth substrate. The PMMA-supported graphene film was then rinsed with deionized water multiple times. Subsequently, the destination substrate (i.e. Si/SiO₂ substrate) was contacted with the PMMA-supported graphene film and the sample was initially dried with nitrogen gun blowing, followed by an oven baking at 80 °C for at least 8 h. Subsequently, the sample was soaked in acetone at room temperature for 6 h to remove the PMMA support layer. The sample was then rinsed with IPA followed by nitrogen blow dry. We note that all process parameters chosen for the PMMA-supported graphene transfer in this work were adopted from earlier reports.^{9,10,16,20,26} A separate experiment was conducted to further verify the advantageous of paraffin support layers over PMMA (Supplementary Note 1).”

“Supplementary Note 1: Modified PMMA-supported graphene transfer process

Here, we conducted a modified PMMA-supported graphene transfer with minimal process variants to that of paraffin, to confirm the advantageous of paraffin over PMMA as a sacrificial support layer. First, a layer of PMMA 950 A5 (Microchem Inc.) was spun at 2,500 rpm for 1 min on the graphene synthesized above Cu foil. The PMMA-coated graphene was then baked in an oven at 80 °C for 1 h. Next, the sample was floated on top of Cu etchant (Copper Etchant TFB, Transense) for 30 min to remove the growth substrate. The PMMA-supported graphene film was then rinsed with deionized water multiple times. Subsequently, the sample was transferred onto 40 °C deionized water surface and kept at the same temperature for at least 1 h. The destination substrate (i.e. Si/SiO₂ substrate) was then used to scoop the

PMMA-supported graphene sample from the 40 °C deionized water from one end. After that, the sample was baked at 40 °C in an oven for more than 24 h. Subsequently, the sample was soaked in a mixture of hexane and acetone (1:1) at room temperature for 12 h to remove the PMMA support layer. The sample was then rinsed with IPA followed by nitrogen blow dry. Supplemental Fig. 7c shows a typical atomic force microscopy (AFM) height profile image of CVD graphene transferred on Si/SiO₂ substrate using the modified PMMA-supported graphene transfer. For direct visual comparison, we reprinted the AFM height profile image of the graphene transferred with conventional PMMA support layer from Fig. 4a in Supplemental Fig. 7a below. As can be seen, the AFM height profile images of graphene transferred with either modified or conventional PMMA-assisted transfer approaches looked comparable in terms of wrinkle density and polymer contamination.

For comparison, we also conducted a modified paraffin-supported graphene transfer experiment where all process parameters were remained the same as the paraffin-supported graphene transfer process described in the Methods (see main text), except for the last sacrificial layer removal step. In this specific experiment, the paraffin support layer was removed using a mixture of hexane and acetone (1:1) at room temperature for 12 h, rather than pure hexane, leaving monolayer graphene on the destination substrate. A typical AFM height profile image of the CVD graphene transferred on Si/SiO₂ substrate using the modified paraffin-supported transfer approach is shown in Supplemental Fig. 7d. As can be clearly seen, the graphene transferred with either standard or modified paraffin support layer shows substantial reduction in terms of wrinkles and polymer contamination.

Overall, the results verify the benefits of paraffin over PMMA, as a sacrificial support layer for graphene transfer technology, in terms of contamination and wrinkle reduction.

Supplemental Fig. 7 | Typical AFM height profile images of graphene film prepared with **a**, PMMA, **b**, paraffin, **c**, modified PMMA, and **d**, modified paraffin support layers transfer method, respectively. Supplemental Fig. 7a, b was reprinted from Fig. 4a, b, for direct visual comparison.

re:"Tg" - Thank you for the new data showing the Tg of the paraffin used here. However, I feel that the point stands that to treat the samples fairly one would dry both PMMA and paraffin at 40°C for 24 hours - especially knowing (as stated in the response) that high temperatures can cause actual damage to the graphene!

Response: Following the reviewer's suggestion, we have dried the PMMA-supported graphene sample at 40 °C in an oven for more than 24 h in the new control experiment, "Modified PMMA-supported graphene transfer process", as detailed in Supplemental Note 1 (reprinted above) for comparison purpose.

re:"hexane vs acetone" - A 50/50 mix of hexane and acetone would dissolve the paraffin and the PMMA over 6 hours and provide the fairest test in my opinion, and both samples could be washed in IPA since all three solvents are freely miscible. The tendency of acetone to be dirty from dissolved plasticisers, self-aldolisation and dissolved water makes me worry that the current hexane-only sample is getting unfair treatment by avoiding acetone despite careful washing.

Response: We thank the reviewer for this comment. We have thus conducted a new experiment: modified paraffin-supported graphene transfer experiment where all process parameters were remained the same as the paraffin-supported graphene transfer process, except for the last sacrificial layer removal step. Following the reviewer's suggestion, in the new experiment, the paraffin support layer was removed using a mixture of hexane and acetone (1:1), rather than pure hexane. Supplemental Fig. 7d shows a typical AFM height profile image of the CVD graphene transferred with the modified paraffin transfer approach. As can be seen, the graphene transferred with this modified paraffin approach still shows substantial reduction in polymer contamination compared to that of PMMA. Supplemental Fig. 7 was reprinted above. Details and results of this new experiment were added as Supplementary Note 1.

re:"PMMA contact during ebeam" - thank you for the changes - there is some evidence that PMMA residues might be relatively benign though, e.g. <https://doi.org/10.1088/2053-1583/1/3/035005>.

Response: We thank the reviewer for this comment and we have added the reference article as reference no. 28 on manuscript page 8.

re:"removing ebeam resist at 60C for 12h" - still, despite the changes, readers might question that the ebeam resist layer is given 12 hours to dissolve, and a somewhat thicker PMMA support layer (judging from the spin speed) only given 6 hours.

Response: We thank the reviewer for this comment. In the new experiment (Supplementary Note 1), we have thus removed both types of sacrificial support layer, *i.e.* PMMA and paraffin, using a solvent (1:1 acetone/hexane) at room temperature for 12 h, rather than 6 h. As can be seen from Supplemental Fig. 7 a and c, both 6 and 12 hours of solvent dipping do not result in observable difference in terms of PMMA residue for graphene samples.

re:"van der Pauw details" - thank you for the description of the method.

re:"paraffin source" - thank you

re:"DFT clarity" - moving this paragraph to the SI has improved readability

T. Booth

We sincerely thank Professor T. Booth for recommending our work for publication in *Nature Communications*. We also deeply appreciate his comments, which helps to improve the robustness of the reported work.

Reviewer #2 (Remarks to the Author):

I thank the authors for their changes to the manuscript, which are very thorough. I believe this work can now be published in Nature Communications.

We sincerely thank the reviewer for supporting our work for publication in *Nature Communications*.

REVIEWERS' COMMENTS:

Reviewer #1 (Remarks to the Author):

I am very glad to see the changes to the manuscript, which I believe make the conclusions very firm - in particular the extensive additional control experiments and employing the 1:1 acetone hexane mixture for support layer removal. I think the control results have implications for the relative importance and performance impact of each part of the process that will be invaluable for interested readers. I thank the authors for their careful attention to my concerns.

T. Booth